# The Human Impact on Changes in the Forest Range of the Silesian Beskids (Western Carpathians)

**Michał Sobala**  **and Oimahmad Rahmonov** *

Faculty of Natural Sciences, University of Silesia, 41200 Sosnowiec, Poland; michal.sobala@us.edu.pl
* Correspondence: oimahmad.rahmonov@us.edu.pl; Tel.: +48-32-3689-306

**Abstract:** Changes in forest range are caused by human activity in many regions of the world. The aim of this paper is an attempt to determine the impact of pastoral and forest management on changes in forest cover and their fragmentation in the Silesian Beskids (southern Poland) in 1848–2015. Historical maps and landscape metrics were used to study changes in forest cover. Using a digital map of forests, analyses of the distribution of forest communities, site types and their condition were conducted. Since 1848 the forest area has increased by 11.8%, while the area of forest core zones has increased by 16.2%, accompanied by a 4.5% reduction in the forest's internal buffer zone. From the mid-nineteenth century, the forest range has been systematically growing from 82.1 to 93.9% because of the pastureland abandonment and forest regeneration, despite temporary logging resulting in forest fragmentation. Minor changes in core area index (CAI) from 80.41 to 87.55 indicate that pastoral economy did not result in considerable fragmentation of forests. The impact of forest management was greater as the sites characterised by natural condition occupy only 28% of the forest land and anthropogenically transformed ones dominate occupying over 50%. An artificial spruce monoculture was died-off and large felling areas were created at the beginning of the twenty-first century covering almost 40% of the study area.

**Keywords:** forest transformation; forest cover; historical maps; pastoral management; land use; landscape changes; Western Carpathians

## 1. Introduction

The degradation of forest cover all over the world is often connected with human activity [1]. The rate of this degradation is different in each climate zone and depends on types of land use [2–5]. In the temperate zone, changes in forest areas in mountain areas were historically connected with sheep grazing and pastoral and agricultural purposes [6–9]. Investigations into the changes in forest ranges have begun within the past 40 years [10–12]. In many papers, forest transformation is considered to be a consequence of land use at the beginning of the seventeenth century. Modern disturbance of forests is strictly connected with the logging of trees for economic purposes, such as industry and charcoal [13,14], as well as with the transformation in many climate zones of forest areas into huge agricultural regions [15–18]. Over the past century, these changes have become more pronounced in all climate zones that support forests and have frequently led to reductions in the extent of forests in favour of agricultural areas and building developments related to rapid population growth in developing countries [19]. Developed countries, on the other hand, have increased their forest ecosystems following a long period of deforestation, especially in marginal areas [20]. Over the past several decades, this has been accompanied by a considerable increase in spatial diversification, causing disruption in the functioning of ecosystems and landscapes [21–23].

Studies of landscape history in North America—unlike those in many regions of Europe, where a wealth of historical data has enabled a rigorous evaluation of landscape history—are

frequently hampered by limited data on past landscape conditions and land-use history. As a result, prior studies in the north-eastern USA have generally been restricted to local investigations of sites with well-documented histories of disturbance [24]. However, in the temperate areas of North America, a pattern has been documented of episodes of deforestation and agriculture followed by agricultural abandonment and widespread natural reforestation [25].

Changes in forest range in the Carpathian Mountains (Central Europe) are related to settlement and the accompanying agricultural and pastoral economy as well as to forest management. Changes in deciduous forests related to pastoralism have been the subject of cartographical studies in Romania [26,27], Slovakia [28,29], Ukraine [30,31], Hungary [17,32,33], and Poland [34,35]. Studies of this type have been conducted throughout considerable areas of the Carpathian Mountains, and sometimes even on an entire segment of the mountain range belonging to a specific country. As a result, they do not always reflect actual changes, as detailed large-scale cartographical studies do. There are only a few papers concerning the changes in forest range occurring within small areas [36]. Moreover, these studies mainly emphasize the impact of agriculture and pastoralism on changes in forest range on the basis of cartographical analyses, but did not analyse the direct impact of forest management causing occurrences of clear-cutting.

In the Polish Carpathians, changes in forest areas have been extensively analysed, but on a small scale and within different time spans [37–39]. The artificial spruce monoculture was introduced into a beech site at the turn of the nineteenth and twentieth centuries. The optimum climate for beech in Europe includes a maximum altitude of 1200 m a.s.l.; thus, beech forests up to this altitude are typical, whereas the introduction of spruce is atypical at such an elevation. The original range of Norway spruce (*Picea abies* (L.) H.Karst) in Europe, in addition to the boreal zone, extends to the montane zone of the Alps and to the Hercynian, Carpathian, Rhodope, and Illyrian regions. Historical information indicates that spruce has also frequently been found at lower altitudes at sites with a permanent high soil-moisture content, and even at sites characterised by a high degree of waterlogging or in peat soils. The present conditions of spruce forests in Central Europe have led to enhanced efforts to reconstruct forests in more stable conditions. In many instances, these conditions can be achieved by means of a "close-to-nature" forest composition, i.e., forests which correspond to the potential natural forest vegetation [40].

Current changes in forest range are being caused by expanded land development, agricultural abandonment, and the development of tourist infrastructure in many regions of the Carpathians [4,6]. Other factors include global climate change, which is considered the major cause of spruce forest death [41], or air pollution, which increases the acidity of soils [42]. This, in turn, decreases the productivity of spruce forest community [43]. The impact of individual factors on changes in forest range varies regionally and depends on the natural and socio-economic relationships that govern various types of human economic activity in specific regions [44,45].

The main types of landscape change across Europe and their driving factors have recently been widely discussed by Plieninger et al. [4]. Similar analyses have been carried out in the Carpathians [46,47], and in areas adjacent to the Beskid Mountains in Poland [48]. These papers were mainly focused on changes in forest cover connected with land abandonment. Changes in forest cover over the past 180 years have been examined in the Western and Northern Carpathians [6]. However, it is not possible to assess fully human impact on forests regarding only changes in forest cover. The aim of this paper is to determine the impact of pastoral and forestry management on changes in forests and their fragmentation in the Silesian Beskids in the period 1848–2015. To this end, we analysed changes in forest cover based on historical maps, and those occurring in contemporary forest communities, as well as site types and their condition based on field study and contemporary cartographical materials.

## 2. Materials and Methods

### 2.1. Study Area

The study area, which is part of the Barania Góra Range in the Silesian Beskid Mountains (Figure 1), covers an area of 45.06 km² and is located within an altitudinal range of 516–1257 m a.s.l. The area is characterised by medium and low mountain reliefs with steep slopes (mean elevation > 800 m a.s.l.). The parent rocks were created from formations of the Godula and Istebna Nappes of the Carpathian Flysch Belt. The areas span three vertical climatic zones, namely, moderately warm (mean temperature > 6 °C), moderately cool (4–6 °C), and cool (<4 °C). Precipitation reaches 1300 mm year⁻¹ on the highest ridges. The natural vegetation consists of fir-beech and fir-spruce forests. Fragments of natural forest are found only on hard-to-reach slopes and in the highest parts of the mountains, with the exception of areas surrounding mountain pastures.

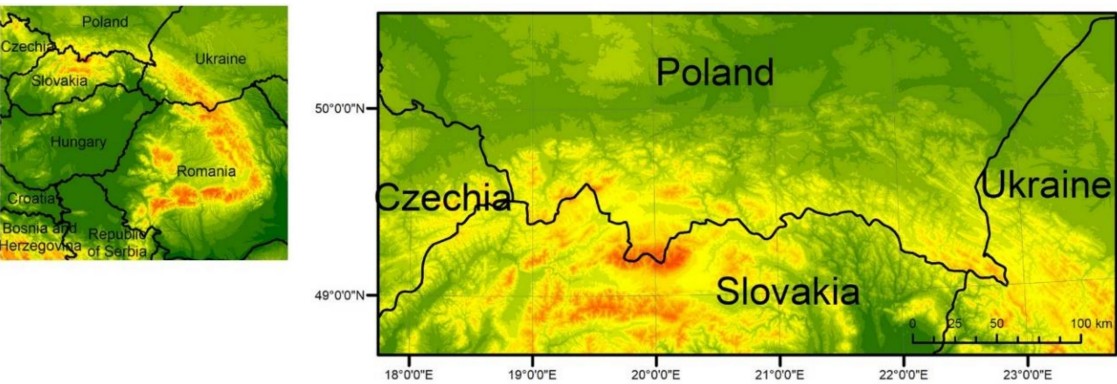

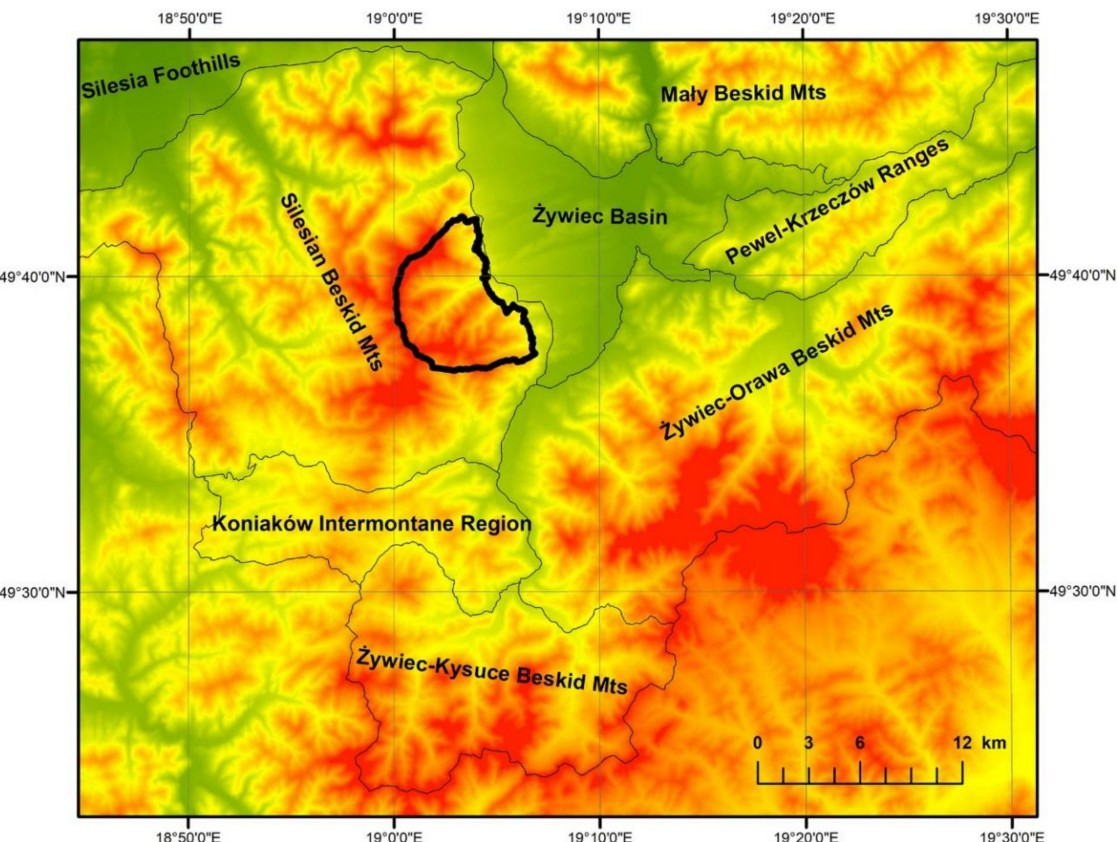

**Figure 1.** Study area.

The altitudinal zonation of the Beskid Mountains (Western Carpathians) and their location in the temperate climate zone have an impact on their natural environment. Forest ranges have undergone considerable changes as a result of human settlement dating back to mediaeval times [35,49]. The beginnings of agriculture in the area date back to the end of the thirteenth century, when settlers migrated to the valley floors. Beginning at the end of the fifteenth century, the arriving Vlachs, who specialised in animal herding, used mountain slopes above 900 m a.s.l., creating large areas of mountain pasture. The area covered by forest was reduced, with the most extensive deforestation occurring in the 1850s. Since then, forest areas have systematically increased [17,26,36]. In recent years, areas which had previously been deforested for pastures have been subject to forest succession, often supported by the afforestation of clearings. Former felling areas have been designated for planned reforestation as well. The study area is the only part of the mountain range which bears traces of the former pastoral economy and is presently subject to forest management.

## 2.2. Spatial Data

The method of historical analysis was used to conduct an analysis of forest cover changes. The following cartographical materials were used:

- Austrian cadastral maps from 1848 scaled at 1:2880;
- A Spezialkarte der Österreichisch-Ungarischen Monarchie (special map of the Austro-Hungarian Monarchy) from 1879 scaled at 1:75,000;
- A WIG (Military Geographical Institute of Poland) military map from 1933 scaled at 1:100,000;
- A military topographic map dated 1960 scaled at 1:25,000;
- A topographic map from 1979 scaled at 1:10,000;
- An orthophotomap dated 2015.

In addition, the following were used: a digital terrain model derived from airborne laser scanning in 2015 (DTM) and a digital map of forests compiled by State Forests, which is a Polish governmental organization that manages Polish state-owned forests. The analysis of forest range based on historical maps was supplemented with written sources (State forest records from the State Archive in Żywiec) and historical studies [50,51].

## 2.3. Cartographical Analysis

Prior to the start of the cartographic analysis, the maps were georeferenced and digitized according to methods presented in the literature on the subject [52–54]. Georeferencing was specifically adjusted to the quality and type of data in order to achieve the best possible results for each series [36]. The processed cartographic materials underwent screen digitalization using the snapping function. The errors usually generated during this operation were eliminated using a topology construction tool [55]. Screen digitisation was combined with the creation of a database of land-cover forms. By aggregating the data included in each series of maps, land-cover maps in which forest and non-forest areas were clearly distinguishable were developed. In both cases, road and hydrologic networks were attached to the adjoining polygons, as the boundaries ran along roads and watercourses. As a result of these procedures, vector maps were created, enabling spatial analyses to take place.

Use of the V_LATE add-on to the ArcGIS package, version 10.5.1., enabled the calculation of landscape metrics describing land cover in each time section. The following parameters were applied:

- Forest area (FA) and non-forest area (NFA);
- Percentages of forest (FP) and non-forest (NFP) areas;
- Forest core area (FCA)—equals the area within the patch that is more than the specified depth-of-edge distance from the patch perimeter; FCA has been found to be a much better predictor of site quality than patch area [56];

- Inner buffer area (IBA)—the area between the edge of the patch and the core area; it was assumed that the buffer zone is 50 m wide [57];
- Percentages of forest core (FCP) and inner buffer (IBP) areas;
- Core area index (CAI)—the percentage of a patch that constitutes the core area;
- Number of forest patches (FNP);
- Number of core areas (CAN);
- Maximal forest patch area (MFA)—the area of the largest forest patch indicated in the study area.

The selection of these landscape metrics was influenced by the susceptibility of a number of phenomena and processes related to the proximity of boundaries [58]. The compactness of forest ecosystems and their boundaries with other ecosystems exert a considerable impact on the way they function [59]. The core zones, i.e., areas within a given ecosystem isolated from the boundaries by buffer zones, are unaffected by, though strictly related to, the edge effect. Due to different environmental conditions, the qualitative and quantitative relationships between components in buffer zones differ from those in the core of a tree stand [57].

Additionally, for the purpose of a detailed analysis of changes in forest extent, three altitudinal zones were distinguished, depending on the type of agricultural activity: below 750 m a.s.l. (permanent settlements and crop cultivation), 750–900 m a.s.l. (pastoral and hay-meadow economy), and above 900 m a.s.l. (pastoral economy). Two types of slope, depending on their inclination, were distinguished: gentle (up to 15°—available for pastoral and hay-meadow economies) and steep (over 15°—reserved for forestation) [37].

The impact of pastoral economy on forest ecosystem was evaluated on the basis of all the archival maps since they all present the range of pasturelands. The impact of forest management was evaluated solely on the basis of the map of the areas of temporarily cleared forests. This map was created on the basis of an orthophotomap dated 2015 and a digital map of forests. The archival maps available for the study area represent only the extent of forest lands and contain no information on felling areas. Hence, it is crucial to distinguish between a forest land and a forest area. A forest area is an area currently covered with forests, whereas a forest land is an area which may temporarily be devoid of forests but it will be reforested in accordance with planned forest management. Therefore, the term forest land is broader than forest area, because it also covers areas that have been temporarily cleared of forest vegetation. As a result, it is impossible to assess the impact of forest management on the basis of archival maps. Hence, a digital map of forests was used and field studies were conducted.

## 2.4. Analyses of Forest Sites and Communities—Field Study

Using a digital map of forests, analyses of the distribution of forest site types (environmental setting) and their condition were conducted. Forest site types include forest areas with similar site features such as soil fertility, diversity and humidity, climate, topographical relief and geology. Areas belonging to the same forest site type are of similar production capacity and usefulness for forest cultivation. The site condition provides information about whether the forest is of natural character or has been anthropogenically transformed. The site condition takes into account all features of site/soil/environment where the forest grows. The site condition was determined by State Forests on the basis of the characteristics of the stand (species composition, forest stratification), ground cover (species composition, coverage), properties of top soil horizons (soil type, humus type and subtype, physical and chemical properties of soil), and soil water conditions. The site condition is determined by comparing the above-mentioned items with elements considered as typical of the area [60].

According to the digital map of forests, the following site conditions were distinguished:

- Natural—unchanged by human activity;
- Close-to-nature—those where the species composition of the stand has been changed because of human activity, but where this change did not influence the ground cover, properties of top soil horizons (mainly humus horizon), or soil water conditions;

- Anthropogenically transformed—those where the species composition of the stand has been changed because of human activity and where this change did influence the ground cover, properties of top soil horizons, and soil water conditions.

Furthermore, field studies that involve verification of the actual range of vegetation were conducted. Firstly, the study area was divided into deciduous and coniferous forests, clear-cutting areas and meadows based on the orthophotomap. In order to identify plant communities, phytosociological releves were taken on individual distinguished surfaces using the Braun-Blanquet method [61]. This method consists of drawing up floristic lists by dividing species into woody, shrubby, and herbaceous ones, and then determining the degree of plant species coverage on particular surfaces. Usually, the phytosociological releve area for forest areas was 400 m$^2$. According to this method, to identify vegetation at the association level, the presence of a statement of characteristic and distinctive species for a given association is sufficient. These species are associated with their very close ecological requirements. Based on the inventoried plant species, the occurrence of species characteristics of mountain forest communities was checked with "A guide for identification of plant communities in Poland", published by Matuszkiewicz [62].

## 3. Results

### 3.1. Changes in Forest Range from 1848 to the Present

Forest cover grew systematically at the expense of nonforest areas in the period 1848–2015 due to the abandonment of agriculture (Figure 2, Table 1). Since 1848, overall forest area has increased by 11.8%, while the area of forest core zones has increased by 16.2%. This has been accompanied by a 4.5% reduction in the forest internal buffer zone, although a slight expansion was recorded over the years 1960–1979. Along with the area of forest, the core zone area index also increased by 7.1%. The greatest changes in forest range, affecting 6.8% of the study area, took place between 1933 and 1960.

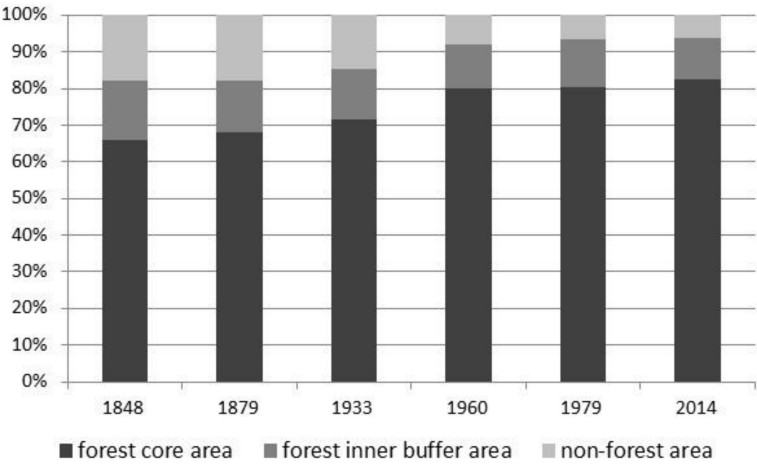

**Figure 2.** Changes in the percentage share of forest and non-forest areas in the years 1848–2015.

**Table 1.** Changes in selected landscape metrics characteristic of forest patch core zones: forest area (FA), non-forest area (NFA), core area index (CAI), number of forest patches (FNP), number of core areas (CAN), maximal forest patch area (MFA).

| Metric | 1848 | 1879 | 1933 | 1960 | 1979 | 2015 |
|---|---|---|---|---|---|---|
| FA (km$^2$) | 36.99 | 37.07 | 38.41 | 41.45 | 42.14 | 42.30 |
| NFA (km$^2$) | 8.07 | 7.99 | 6.65 | 3.61 | 2.92 | 2.76 |
| CAI | 80.41 | 82.73 | 83.83 | 86.73 | 85.91 | 87.55 |
| FNP | 9 | 2 | 1 | 12 | 13 | 18 |
| CAN | 19 | 7 | 9 | 20 | 26 | 28 |
| MFA (km$^2$) | 36.94 | 36.99 | 38.41 | 41.08 | 42.09 | 42.26 |

Within the entire analysed time span, the number of forest patches and their core zones increased. Throughout 1848–2015, the number of forest patches was lower than that of the core areas. Additionally, the area of the largest forest patch increased by 5.3 km².

The share of forest areas throughout the entire analysed time span increased with altitude and slope inclination (Figure 3). At the same time, the most dynamic expansion of forest area occurred in the lowest altitudinal zone and on slopes with an inclination below 15°, as opposed to steep slopes. Although the general tendency favours increases in forest area, it is also important to note that slight decreases in forest range occurred in the years 1979–2015 on steep slopes in the altitudinal zone 750–900 m a.s.l. and in the years 1848–1879 in the altitudinal zone above 900 m a.s.l.

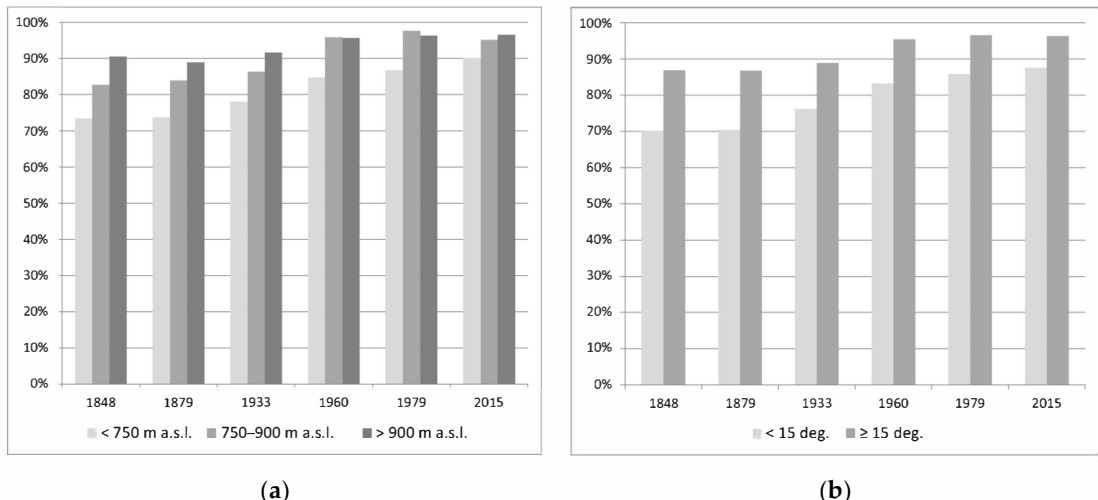

(**a**)　　　　　　　　　　　　　　　　　　　　　　　(**b**)

**Figure 3.** Changes in the percentage share of forest area in the years 1848–2015: (**a**) in different altitudinal zones; (**b**) in slopes with various inclinations.

The results listed above apply to forest range as presented on the maps, which in fact represents the range of forest land and does not account for felling performed as part of forest management. Archival maps available for the study area contain no information on felling areas; thus, they can be analysed only in terms of the present, based on field verification of forest range and on the orthophotomap. When comparing the range of forest land represented on maps to the actual (2015), it can be observed that the percentage share of the actual forest area is currently 36.2% lower than the share of forest land (Figure 4). This includes a higher actual share for the forest buffer zone area (by 7.8%) and a lower share for the core zone area (by 43.9%) (Table 2). Another effect of including felling areas in the analysis is that the number of forest patches is considerably greater and the area of the largest separate forest patch is smaller.

**Table 2.** Comparison of selected landscape metrics characteristic of core zones in forests and forest areas: percentages of forest areas (FP), inner buffer areas (IBP) forest core areas (FCP) and non-forest areas (NFP), core area index (CAI), number of forest patches (FNP), number of core areas (CAN), maximal forest patch area (MFA).

| Metric | Forest Areas Excluding Felling Area (Forest Land) | Forest Areas Including Felling Area (Forests) |
|---|---|---|
| FP (%) | 93.9 | 57.7 |
| IBP (%) | 11.5 | 19.3 |
| FCP (%) | 82.3 | 38.4 |
| NFP (%) | 6.1 | 42.4 |
| CAI | 87.55 | 63.90 |
| FNP | 18 | 114 |
| CAN | 28 | 225 |
| MFA (km²) | 42.26 | 25.50 |

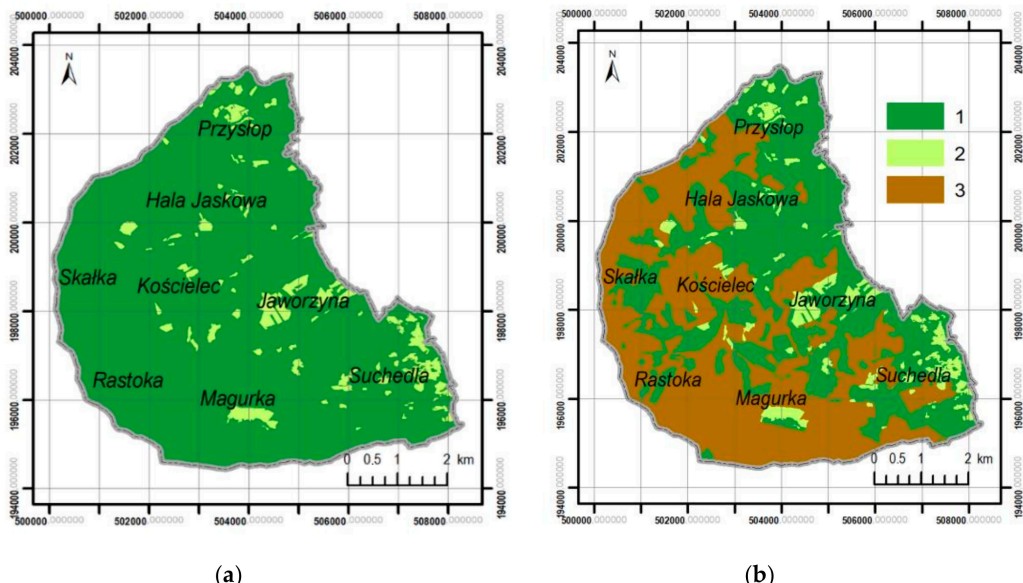

**Figure 4.** Current range of forest areas: (**a**) excluding felling areas; (**b**) including felling areas. 1—forest lands, 2—non-forest areas, 3—felling areas.

### 3.2. Differentiation of Contemporary Forest Ecosystems

According to the typology of forest sites conducted by the Polish State Forests, two types of sites dominate in the study area: mixed mountain forests on fresh sites (65%) and mountain coniferous forests on fresh sites (26%) (Figure 5). Furthermore, small areas are covered by the following sites: mountain broadleaf forest on fresh sites (7%) and mountain riparian forests. As a result of our field studies, based on the presence of characteristic species for individual communities, the following natural and close-to-nature communities were found in the Silesian Beskids: submontane riparian forests (*Caric remotae-Fraxinetum*) and grey alder forests (*Alnetum incanae*) that are typical of mountain riparian forest sites, the Carpathian beech forests (*Dentario glandulosae-Fagetum* and *Luzulo nemorosae-Fagetum*) that are typical of mountain broadleaf forest on fresh sites and mixed mountain forests on fresh sites, and the Carpathian fir-spruce forests (*Abieti-Piceetum montanum*) and Carpathian spruce forests (*Plagiothecio-Piceetum tatricum*) that are typical of mountain coniferous forests on fresh sites. All of these communities occur in small areas which are difficult to cultivate.

The stand of *Abieti-Piceetum montanum* consists of Norway spruce (*Picea abies*) with a modest occurrence of European beech (*Fagus sylvatica* L.) and European silver fir (*Abies alba* Mill.). Apart from tree saplings, the shrub layer comprises mountain-ash (*Sorbus aucuparia* L.) and, rarely, black-berried honeysuckle (*Lonicera nigra* L.). The undergrowth comprises mostly European blueberry (*Vaccinium myrtillus* L.), wavy hair-grass (*Deschampsia flexuosa* L.), small reed (*Calamagrostis arundinacea* L. Roth.), and broad buckler-fern (*Dryopteris dilatata* (Hoffm.) A. Gray)). Similarly, the area of *Plagiothecio-Piceetum tatricum* has been reduced to small patches formed by *P. abies* and *S. aucuparia* found in the upper parts of the mountains. In this case, the shrub layer is absent or formed by saplings of spruce and mountain-ash. The ground cover comprises *V. myrtillus*, alpine lady-fern (*Athyrium distentifolium* Tausch ex Opiz, 1820), and small reed (*Calamagrostis villosa* (Chaix) J. F. Gmel).

In addition, a limited number of forest communities typical of this climate zone (montane zone) often occupy steep, hard-to-reach slopes. These communities are usually characterised by a nearly complete species composition. *Dentario glandulosae-Fagetum* comprises *F. sylvatica*, sycamore maple (*Acer pseudoplatanus* L.), European ash (*Fraxinus excelsior* L.), *A. alba*, and *P. abies*. The shrub layer, formed from saplings, is very poorly developed or completely absent. The undergrowth is very well developed and varied, consisting mostly of coralroot (*Dentaria glandulosa* Waldst. & Kit.) and Braun's holly fern (*Polystichum braunii* L.). Finally, *Luzulo nemorosae-Fagetum* comprises *F. sylvatica*

with some *P. abies* and, sometimes, *A. pseudoplatanus*. The shrub layer is formed from saplings and, like the undergrowth layer, is very diverse. There are also small parts of forest communities related to non-autogenous soils along larger river valleys, i.e., *Alnetum incanae* and *Carici remotae-Fraxinetum*.

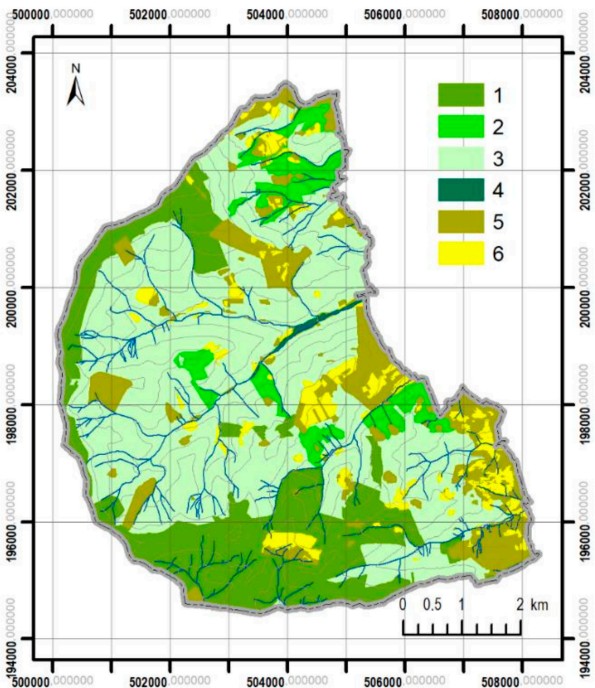

**Figure 5.** Forest site types: 1—mountain coniferous forests on fresh sites, 2—mixed mountain forests on fresh sites, 3—mountain broadleaf forest on fresh site, 4—mountain riparian forests, 5—forest regeneration in abandoned farmlands, 6—meadows.

Nevertheless, the study area is covered mainly by spruce monocultures (50–70% in the Silesian Beskids)—78% of mixed mountain forests on fresh sites are anthropogenically changed (Table 3). In contrast to the forest species composition mentioned above, spruce monocultures are of secondary origin (average age is 70 years) characterised by a very limited species composition. There are single acidophilic species in the ground cover or it is often completely devoid of species due to the lack of light access to the forest floor. Spruce monocultures are artificial stands with little similarity to *Abieti-Piceetum montanum*, characterised by the presence of *P. abies* saplings in the shrub layer or the complete absence of this layer. In recent years, spruce monocultures have been dying out as a result of prolonged weakening processes caused by natural (large-scale infestation with bark beetles) and anthropogenic (air pollution, incompatibility with sites) factors. This process occurred on almost 40% of the study area (Figure 4b). Clear-cuttings were formed on almost 90% of the mountain coniferous forests on fresh sites and almost 50% of mixed mountain forests on fresh sites (Table 4). They were least related to mountain broadleaf forest on fresh sites and they did not occur on mountain riparian forest sites. Clear-cuttings occurred to the greatest extent on close-to-nature sites.

As far as forest sites are concerned, anthropogenically transformed ones dominate on the study area. They occupy over 50% of forest lands (Figure 6, Table 3). The mountain coniferous forests on fresh sites have been transformed the least. They are located in the highest parts of the study area and due to inaccessibility, they have only been slightly transformed by forest management. The sites of mountain riparian forests have been preserved in their natural state. They constitute a small patch fringing the stream in Leśna valley, which is irrelevant from an economic point of view. In admixtures, in forests (especially in mid-forest clearings due to light accessibility), regeneration of F. sylvatica and A. alba was observed, indicating the direction of the renaturalization of the forest.

**Table 3.** Forest site condition. Site types: MCF—mountain coniferous forests on fresh sites, MBF—mountain broadleaf forest on fresh site, MMF—mixed mountain forests on fresh sites, MRF—mountain riparian forests. Site condition: N1—natural, N2—close-to-nature, A—anthropogenically transformed.

| Site Type | Area | | Area of Sites in Different Conditions | | | | | |
|---|---|---|---|---|---|---|---|---|
| | (ha) | (%) | (ha) | | | (%) | | |
| | | | N1 | N2 | A | N1 | N2 | A |
| MCF | 968.2 | 26.6 | 439.9 | 458.7 | 15.6 | 51.0 | 47.4 | 1.6 |
| MBF | 265.0 | 7.3 | 119.2 | 39.3 | 106.5 | 45.0 | 14.8 | 40.2 |
| MMF | 2378.3 | 65.3 | 397.6 | 125.9 | 1854.8 | 16.7 | 5.3 | 78.0 |
| MRF | 12.9 | 0.4 | 12.9 | 0.0 | 0.0 | 100.0 | 0.0 | 0.0 |
| Total | 3624.4 | 100.0 | 1023.5 | 623.9 | 1976.9 | 28.2 | 17.2 | 54.6 |

**Table 4.** Percentage of the clear-cutting areas within particular sites.

| Feature | Percentage of the Clear-Cutting Areas |
|---|---|
| **Site Type** | |
| Mountain coniferous forests on fresh sites | 87.5 |
| Mountain broadleaf forests in fresh sites | 8.1 |
| Mixed mountain forests on fresh sites | 44.3 |
| Mountain riparian forests | 0.0 |
| **Site Condition** | |
| Natural | 47.9 |
| Close-to-nature | 84.0 |
| Anthropogenically transformed | 45.9 |

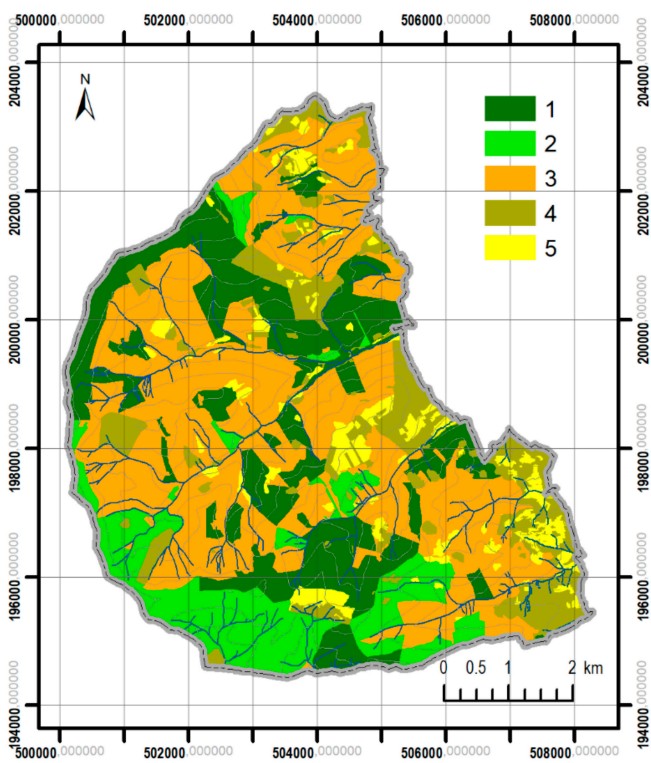

**Figure 6.** Forest site condition: 1—natural sites, 2—close-to-nature sites, 3—anthropogenically transformed sites, 4—forest regeneration in abandoned farmlands, 5—meadows.

## 4. Discussion

Humans have been transforming the earth for centuries; throughout the last half-century, the rate of transformation of natural ecosystems and cultural landscapes has been much greater than ever before. Even in the historical past, social, economic, and environmental problems exerted considerable influence on both local and global scales [16,21].

Our results show that forest cover changes in the Silesian Beskids can be split into four stages (Figure 7). The first stage, involving the reduction in the forest area in the Silesian Beskids, began in the fifteenth century and continued until the nineteenth century [63]. This stage cannot be presented on maps and supported by statistical data because it concerns the time before any cartographical materials had been completed. The level of deforestation reached its peak in the nineteenth century. Deforestation was caused mostly by human settlement extending ever higher up the valley slopes, accompanied by agricultural and especially pastoral activities [13]. Initially, changes occurred over a small area; however, as time progressed, their range increased, as reflected in the maps from the end of the eighteenth century. The increase in mountain grazing led to the development of semi-natural non-forest communities often located in the central parts of forests. The seventeenth century is known as "the golden period" of mountain grazing as a result of the significant increase in headage. Such changes also took place in Czechia [64,65], Hungary [17], Ukraine [66], and Slovakia [12].

Since the nineteenth century (the second stage), an inconsiderable but gradual increase in the forest range has been visible (Figure 7). This was caused by a mountain grazing collapse connected with forest management intensification, the abolition of serfdom, the devaluation of sheep products, and further expansion of built-up areas and arable lands. Subsequently, an increase in forest range at the turn of the nineteenth and twentieth centuries was caused by an exodus of people seeking work in industrial centres and beneficial conditions for farming in the lower parts of the mountains (below 600 m a.s.l.) [50]. As was proven, the most dynamic expansion of the forest area occurred in the lowest altitudinal zone (up to 750 m a.s.l.) and on slopes with an inclination below 15°. This was connected with the fact that in that part of the study area the farmlands covered the greatest surface. In abandoned areas in the upper parts of the mountains, the forest grew back in the course of natural succession, resulting in an increase in the area of forest ecosystems throughout the Carpathian Mountains [35,36,46,47]. Management of the forest was connected with the introduction of spruce monocultures and applying a clear-cutting method. In the Inter-War period (1918–1939), a relative stability of mountain grazing was observed as well as a restriction in forest clearings [51].

Minor changes in core area index (CAI) indicate that pastoral economy, settlements, and crop-growing pressure from the valleys did not result in considerable fragmentation of forests in the Silesian Beskids, as was the case in other parts of Poland and Europe [67,68]. Even though the boundary between farmland and forest moved up the slopes, as is also characteristic of other parts of the Carpathian Mountains [37], forest ranges remained compact. Fragmentation of forests in this area was mostly caused by the felling of high-grade trees suitable for industrial use. A comparison of selected indicators characterising core zones in forests and forest areas shows that in recent years clear-cutting has had a great influence on the current forest fragmentation (Tables 1 and 2). It may be surmised that there was a similar situation in the nineteenth century. However, it is impossible to assess this because of a lack of spatial data in the intervening years. Moreover, this is not to say that the agricultural use of the area had no impact on the functioning of forest ecosystems [69]. In forests surrounding villages, trees were felled, brushwood collected, bark and inner bark removed from trees, leaf litter collected for animal feed, deciduous tree saplings cut down, animals grazed, undergrowth collected, and forests used for hunting [49].

The gradual withdrawal from pastoralism and farming which began in the second half of the nineteenth century had a positive impact on the functioning of forest cover through the introduction of new species and the formation of innovative ecological niches [59,70]. Ecoclines appeared between forest ecosystems and farming areas which had previously been separated by straight-line boundaries. As a result of ongoing forest regeneration, a vast ecotone zone of irregular shape was created between

the two. This kind of border reduces the negative impact of open communities on the forest and enhances biological diversity through the creation of a site suitable for various species adapted to living in transient conditions.

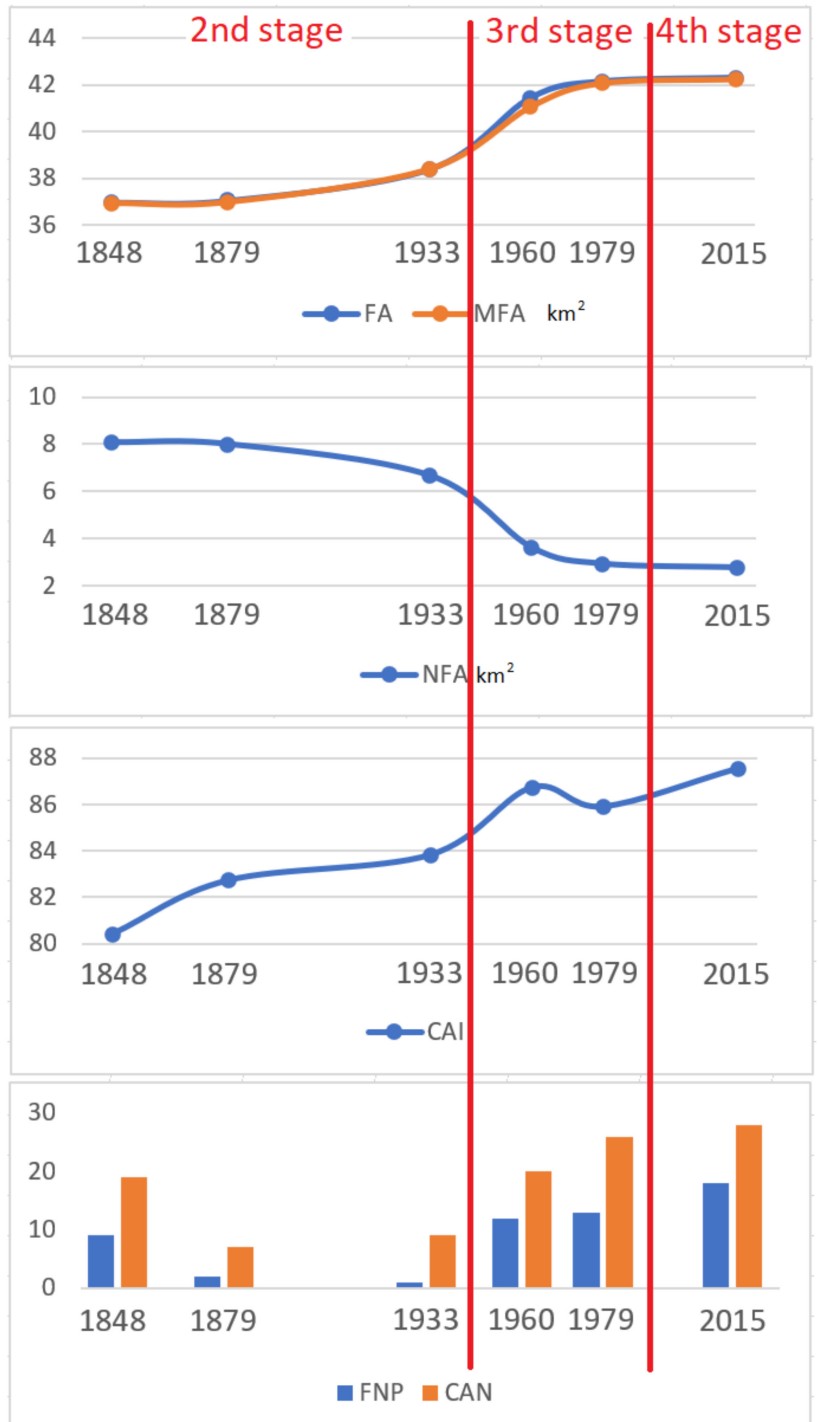

**Figure 7.** Stages of change in forest cover against the background of the landscape metrics: forest area (FA), maximal forest patch area (MFA), non-forest area (NFA), core area index (CAI), number of forest patches (FNP), number of core areas (CAN).

The third stage of changes in the forest cover started after the Second World War (Figure 7). These are visible in the acceleration of change in the forest area, non-forest area, maximal forest patch area and number of forest patches. Further mountain grazing collapsed causing forest regeneration

in the abandoned farmland and an increase in the forest range. Forest stand rebuilding started and continued during that stage, consisting of the introduction of species typical of the area, such as fir and beech. At the turn of 1980s and 1990s, mountain grazing was totally removed. As a result, mountain pastures formerly used for pastoral purposes were subject to forest regeneration, accompanied by a reduction in biological diversity within these areas [71].

The fourth stage started at the turn of the twentieth and twenty-first centuries with a restoration of mountain grazing for the purpose of nature conservation. As a result, reforestation was cramped, which is especially visible in a stabilization in the extent of the forest area, non-forest area and maximal forest patch area (Figure 7). Currently, there is widespread demand for the retention of forest clearings, which are not only important for preserving protected plant and animal species and landscapes, but also for shaping local identity, tradition, and folk culture, stimulating local entrepreneurship, and enhancing the region's tourist values. A number of activities related to the protection of non-forest areas are being conducted in the study area, financed with EU funds (e.g., the LIFE+ programme) and from other sources, with the aim of preserving the open space of mountain pastures and forest clearings through the traditional grazing of sheep, especially in locations where communities (*Hieracio-nardetum*) and species (*Campanula serrata* (Kit. ex Schult.) Hendrych) with community priority are present [9].

As was mentioned above, archival maps contain information only on the range of forest lands, not on the actual presence of forests. Thereby, the analysis of forest range based on historical maps was supplemented with written sources, which indicated that considerable deforestation also occurred in the nineteenth century [51]. This process was related to the response of forestry to industry's growing demand for wood in many regions in the Carpathians [72,73]. The clear-cutting method was used. Selection thinning was also used in order to improve or regenerate a stand. Furthermore, it must be emphasised that the data gained from historical maps are related to mapping time horizons, but multiple or repetitive changes between the mapping horizons remain unknown [74].

Large-scale exploitation of forest ecosystems in the nineteenth century resulted in the creation of clear and group felling areas, which were later subject to natural regeneration and reforestation through planting. Moreover, afforestation involved mostly *P. abies* and was performed in the montane zone, in which nearly all of the Silesian Beskids is located. Undoubtedly, forest management made a considerable impact on the environmental aspects of the sites, changes in species composition within the undergrowth, and general relationships within ecosystems [57,58]. As we showed, anthropogenically transformed sites dominate in the study area (over 50% of forest lands). This is connected mainly with the introduction of spruce monocultures, but it must be emphasised that the site condition depends on the location of this type of forest. Spruce monocultures introduced in place of *Abieti-Piceetum* or *Plagiothecio-Piceetum* can be treated as located in close-to-nature sites. In this case, the forest stand is not completely compatible with the natural forest stand because of the introduction of *P. abies* and the elimination of *F. sylvaticum*, *A. pseudoplatanus* and *A. alba*. However, because coniferous species used to dominate in the above mentioned communities, the elimination of an admixture of deciduous species did not change the properties of top soil horizons or ground cover. As a result, the existing stand is not fully typical of a fresh site while soil conditions fully meet the criteria granted for the close-to-nature sites. In turn, spruce monocultures introduced in place of *Dentario glandulosae-Fagetum* and *Luzulo nemorosae-Fagetum* can be treated as located in anthropogenically transformed sites. This is because of the fact that the introduction of *P. abies* was connected with the elimination of dominant species such as *F. sylvatica*. Such huge changes in stand had an influence on changes in the properties of top soil horizons and ground cover [75–78].

Artificially created spruce monocultures are more susceptible to both anthropogenic and natural influences [79]. As opposed to natural communities, they are more often subject to wind snap or windfall. Indirectly, they contribute to increased flood risk and soil erosion. In turn, natural forest communities provide more ecosystem services which can be priced [80]. Spruce stands have been dying out since the end of the 1950s. The dying-off process in mountain spruce forests was caused by several biotic (bark beetles) and non-biotic factors (air pollution), forming a cause-and-effect chain

that reinforced the disease process, resulting in a total annihilation of the stand over an extended period of time [81–83]. Air pollution, prolonged summer droughts in the years 2003–04 and 2006–07, and the catastrophic hurricane of 2004 resulted in the weakening of spruce stands in the montane zone [84,85]. Bark beetles attacked the weakened trees as part of a large-scale infestation which affected a considerable forest area in the Silesian Beskids within a short time [86]. Spruce monocultures in large areas were degraded to such an extent that tree clearance appeared necessary (Figure 4b).

Over the past few decades, the re-naturalisation of forest ecosystems implemented in the Carpathians has consisted of the introduction of species typical for the area, such as fir and beech. This strategy, adopted by the administration of state-owned forests in Poland in response to the requirements implemented throughout the European Union, will lead to the stability of the relevant ecological systems [87]. Beech forests will probably remain the most important natural forests in temperate Europe during this century [88]. The present conditions of spruce forests in Central Europe have led to enhanced efforts to reconstruct forests so as to create more stable and resilient conditions, which, in many instances, may be achieved by a close-to-nature forest composition, i.e., one corresponding to potential natural forest vegetation and sites [40]. The problem of forest re-naturalisation also applies to North America. Shifley et al. [89] emphasise forest age, structure and vegetation diversity as they relate to changes in wildlife diversity, ecosystem resilience, ecosystem services, and associated forest attributes. Increasing forest age-class diversity should increase other measures of diversity as well as resilience in the face of many types of future forest disturbances. The plant species inventoried in the study area do not differ from those described for forest communities in other parts of the Carpathians [90]. This confirms the ongoing renaturalization of forest ecosystems in the Carpathians. Phytosociological studies can be the basis for the statement that, despite prolonged anthropopressure, close-to-nature forest communities still cover large parts of the Carpathians and have a chance of complete regeneration [91].

A floristic-phytosociological analysis carried out in the area of distinguished forest communities confirms the development of sowing of natural forest ecosystems. The plant species we found in this area do not differ from the described forest communities in other parts of the Carpathians [46]. Phytosociological studies can be the basis for the statement that, despite long-term anthropopressure, the discussed forest communities are in almost full ecological balance. Contemporary pastoralism is small-scale and in no way affects changes in forest vegetation. Thus, in the near future forest ecosystems may achieve complete ecological stability in terms of species composition. Similar forest landscapes in other parts of the Carpathians with a similar species composition have been protected. Against the background of global changes and taking into account the demand for wood, it can be stated that forest ecosystems in this area have a chance of complete regeneration.

As our results show, clear-cuttings have occurred to the greatest extent on close-to-nature sites. This is because of the fact that clear-cuttings have occurred mainly within spruce monocultures introduced in *Abieti-Piceetum* or *Plagiothecio-Piceetum* located in the higher parts of the study area. These are forests in close-to-nature sites because the introduction of *P. abies* made this forest stand not completely compatible with the natural forest stand, but the ground cover, properties of top soil horizons, and soil water conditions are compatible with the natural state. Artificial domination of only one tree species (*P. abies*) caused the stand to break down. In turn, anthropogenically transformed sites in the study area are located in lower parts of the mountain, in valleys and on gentle slopes. This location of anthropogenically transformed sites results from the availability of these areas for forest management—logging in these areas was easier and therefore economically justified compared to the higher parts of the study area. *Dentario glandulosae-Fagetum* and *Luzulo nemorosae-Fagetum* are typical of these sites. The introduction of *P. abies* in these sites caused not only a change in the species composition, but also a change in soil and undergrowth properties [92]. However, apart from spruce, there are also other species of trees: *F. sylvatica* and *A. alba*. This species diversity meant that the scope of tree felling was not as large as in the case of the higher areas, where the sites are close-to-nature, but the stand was characterized by the occurrence of only one species: *P. abies* [81,93].

Because clear-cuttings have appeared in recent years on almost 90% of the mountain coniferous forests on fresh sites located in the highest parts of study areas, the changes in forest range in the Carpathian Mountains may be affected by global climate change [94,95]. Thermal conditions exert a considerable impact on the growth of beech stands, and thus these changes may result in beech becoming the major species in upper parts of the mountains. If global warming continues, beech may retain its production capacity in warm and fresh sites and increase it in cooler areas [96]. Nevertheless, following the destruction of spruce monocultures, spruce is the most dominant regeneration species in terms of the number of specimens at the highest elevations. Similarly, in the forests of North America, climate change may make the future spatial distribution of tree species substantially different from the current distribution [15,97]. Analyses modelling short- and long-term forest changes associated with alternative climate scenarios indicate changes in forest ecosystems in North America [98] and Asia [99–101]. As pointed out by Frelich and Reich [102], the warming climate will have major effects on boreal and northern hardwood forests situated near the prairie–forest border of central North America within the next 50–100 years. A similar situation led to periods of drought in the Carpathians, contributing to the weakening of spruce and causing beech to regenerate naturally [103]. The climate of the future will likely lead to higher mortality among mature trees because of the greater frequency of droughts, fires, forest-levelling windstorms, and outbreaks of native and exotic insect pests and diseases.

## 5. Conclusions

The conducted research allows the following conclusions to be drawn:

1. From the mid-nineteenth century, the forest range in the Silesian Beskids has been systematically growing, from 82.1 to 93.9%. This is related to the gradual abandonment of the pastoral economy and forest regeneration, despite temporary logging resulting in forest fragmentation.
2. Pastoral economy, settlements, and crop-growing pressure from the valleys did not result in a considerable fragmentation of the forests in the Silesian Beskids.
3. The lack of cartographic data showing the temporary surfaces of deforestation related to forest management made it impossible to conduct a quantitative analysis of the impact of forest management on forest ecosystems fragmentation in the past. Nevertheless, it can be assumed that the impact of forest management on the condition of forests was greater than that of pastoral economy. This was reflected by changes in the species compositions of forest stands in the last 150 years. The preference for spruce resulted in the replacement of natural ecosystems with spruce monocultures. As a consequence, the massive dying-off of this type of forest was observed and large felling areas were created at the beginning of the twenty-first century covering almost 40% of the study area.

Using historical maps, it is possible to determine only changes in the forest cover. However, it is insufficient to assess human impact on forests as archival maps very often contain information on the range of the forest lands only, not on the actual presence of forests and their characteristics. Therefore, in performing analyses of this kind, it is important to distinguish between forest lands and forests. Furthermore, forest communities, a range of temporary clear-cutting connected with forest management, site types and their condition should be also analysed to assess fully human impact on forests.

**Author Contributions:** O.R. and M.S. wrote the main manuscript. O.R. conducted field studies. M.S. conducted GIS analysis. All authors have read and agreed to the published version of the manuscript.

**Funding:** This research received no external funding.

**Acknowledgments:** We gratefully acknowledge two anonymous reviewers for their constructive comments.

**Conflicts of Interest:** The authors declare no conflict of interest.

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
