# Peer review of "The Human Impact on Changes in the Forest Range of the Silesian Beskids (Western Carpathians)"

_resources, doi:10.3390/resources9120141_

Round 1

Reviewer 1 Report

The manuscript seems to be interesting and it fits the scope of the journal. But, in submitted form, the manuscript has many shortcomings.

Firstly, the title of the manuscript is a bit confusing. Authors deal with land-use historical changes in study area as main topic of the study. Land-cover is not usually considered as “forest resource”. This terminological problem must be seriously reworked in the whole body of manuscript (including title). By the way, Authors did not precisely define this key term of the study (it is missing in lines 9 and 27, at least).

Secondly, some important shortcomings are in section Methods and consequently in section Results (see my detailed comments below). I suggested major reworking some parts of the manuscript – removing of subchapters 2.4 and 3.2. This study can be presented as well-written land-use analyses in study area, it should be enough for good scientific paper.

Thirdly, some missing important citations (strictly related to the topic of the study) should be added to references and discussed in section Discussion: doi:10.1016/j.apgeog.2016.12.016; doi:10.3897/natureconservation.22.12902; doi: 10.3390/f8110427.  

Detailed comments:

Line 9, 27: I recommend to Authors reworking the main idea of the study. It should be better to avoid using the term “forest resource” and beyond it to use the correct terms “forest land-use” or “forest cover” in the study.

Line 10: Not “forestry management” but “forest management” is correct term.

Line 12: Only one sentence for methodology presentation is too poor. Authors should add two – three sentences about methodology.

Line 18: What means “probably greater”? Authors should avoid using unclear statements.

Line 19: There is also unclear what means the term “natural conditions” in this context. It is not clear in Methods also.

Line 22: Norway-spruce monoculture decline is obviously the most human impact in this study area. Authors should discuss it seriously in section Discussion, but not present it as “results”, because of this topic did not study in the paper.

Line 27: This statement is not true. Economical issues can support sustainable forest management (see my recommendation add some ignored literature above).

Line 81-83: Authors must explain briefly what novelty this study deals with in comparison to cited studies no 46-47.

Line 83: This deals not with “agricultural usage”.

Lines 83-86: Main aim of the study must be seriously reworked (see my comments to the title and main idea of the study). Also any hypotheses in tested must be added here.

Line 89-111: Description of study area must be brief and related to the topic. Any information related to forest history and historical land-use changes must be added.

Line 127-179: This section is too much similar as part of theses. It must be re-structuralised and presented more briefly, avoiding using well-known methodological steps.

Line 180-211: This methodological approach seems not to be professional. Categories in lines 192-199 seem to be very simply and very disputable. Also, field recognition of these categories in the field, presented in this study, is too trivial without professional access. I strictly suggest to Authors remove this part of the study from the manuscript and concentrate only to serious presentation of results on land-use long term changes in study area.  

Lines 255-323: This part should be removed by my opinion. It consists of not very professional applying of methodological approach.

Lines 501-502: Authors should aware to present only real results here – this study did not investigate “impact of pastoral economy and forest management on…”.

Lines 501-533: This section should be reworked: Authors should avoid presentation of general statements here. Only very brief and concise summary of main original scientific findings of the study should be clearly presented here. Authors should emphasise especially such results, which are novelty and which fill some knowledge-gaps mentioned in section Introduction. Also, it is important that Authors should emphasise here some internationaly important results of the study, which can be interesting for international audience of the journal. Are there any results, which are portable to other areas and which support sustainable using of natural resources such as land, forests etc?

Final note: I suggest to Authors consider using soma statistical methods for estimation of any human impacts to land-use changes in study area. It should be fine using e.g. regression analyses in order to evaluate impacts of some human activities if these could be defined as variables…

Reviewer 2 Report

The human impact on changes in the forest resources of the Silesian Beskids (Western Carpathians)

I enjoyed reading the manuscript thoroughly and found that authors have done an excellent job to determine the impact of pastoral and forestry management on changes in forest range and their fragmentation in the Silesian Beskids (southern Poland) in 1848– 12 2015. In overall, the manuscript is well written in a clear and understandable language. However, authors need to give some attention in maintaining the flow of the message as well as English editing issues (spelling/ tense..).

Specific:

Abstract

Abstract is clear. Nicely captured. However, additional one or two sentences on methodology and implications of the study findings will enhance clarity and quality of the paper.

Introduction

Introduction needs improvement in flow of the messages. Authors need to elaborate further on the knowledge gap in the topic as well as the rationale of this research.  

Materials and Methods

Excellent presentation. Methods described in the manuscript is written well and scientifically logical. However, I am not much confident to comment on the cartographical analysis aspect. Hope other reviewer can contribute.

Results

Straight-forward and presented nicely.

Discussion

Discussion section is quite comprehensive and written clearly. However, some of the aspects presented in the result sections are not discussed thoroughly in the discussions. For example, I could not find the reasons behind increase of forest resources with slope and altitude (presented in result section) discussed in this section.

Conclusions

Well written but better if author could shorten the conclusions.

Round 2

Reviewer 1 Report

Dear Authors, thanks for your replay and corrections of the submitted manuscript.